# The topography of frequency and time representation in primate auditory cortices

Simon Baumann[1]*, Olivier Joly[1,2], Adrian Rees[1], Christopher I Petkov[1], Li Sun[1], Alexander Thiele[1], Timothy D Griffiths[1]

[1]Institute of Neuroscience, Newcastle University, Newcastle upon Tyne, United Kingdom; [2]MRC Cognition and Brain Sciences Unit, Department of Experimental Psychology, University of Oxford, Oxford, United Kingdom

**Abstract** Natural sounds can be characterised by their spectral content and temporal modulation, but how the brain is organized to analyse these two critical sound dimensions remains uncertain. Using functional magnetic resonance imaging, we demonstrate a topographical representation of amplitude modulation rate in the auditory cortex of awake macaques. The representation of this temporal dimension is organized in approximately concentric bands of equal rates across the superior temporal plane in both hemispheres, progressing from high rates in the posterior core to low rates in the anterior core and lateral belt cortex. In A1 the resulting gradient of modulation rate runs approximately perpendicular to the axis of the tonotopic gradient, suggesting an orthogonal organisation of spectral and temporal sound dimensions. In auditory belt areas this relationship is more complex. The data suggest a continuous representation of modulation rate across several physiological areas, in contradistinction to a separate representation of frequency within each area.

## Introduction

Frequency structure (spectral composition) and temporal modulation rate are fundamental dimensions of natural sounds. The topographical representation of frequency (tonotopy) is a well-established organisational principle of the auditory system. In mammals, tonotopy is established in the receptor organ, the cochlea, and maintained as a systematic spatial separation of different frequencies in different areas of the ascending auditory pathway, and in the auditory cortex. While temporal modulation is recognised as an essential perceptual component of communication sounds such as human speech and animal vocalisations (*Rosen, 1992*; *Drullman et al., 1994*; *Shannon et al., 1995*; *Wang, 2000*; *Chi et al., 2005*; *Elliott and Theunissen, 2009*), its representation in the auditory system is poorly understood. In contrast to sound frequency, amplitude modulation rate is not spatially organised in the cochlea but represented in the temporal dynamics of neuronal firing patterns. However, a considerable proportion of neurons in the auditory brainstem and cortex show tuning to amplitude modulation rates (reviewed in *Joris et al., 2004*). It has been proposed that temporal information in sound is extracted by amplitude modulation filter banks (*Dau et al., 1997a, 1997b*) that are physiologically instantiated in the midbrain (*Rees and Langner, 2005*). Studies in rodents (*Langner et al., 2002*), cats (*Schreiner and Langner, 1988*) and primates (*Baumann et al., 2011*) have shown that at the stage of the inferior colliculus amplitude modulation rate and frequency of sound are represented in approximately orthogonal topographical maps.

Whether the spatial organisation of amplitude modulation rate is preserved in the auditory cortex remains is debated. Data from gerbils (*Schulze et al., 2002*) and cats (*Langner et al., 2009*) suggest a topographical map for temporal modulation rates in the auditory cortex. In cats, orthogonal

*For correspondence: simon.baumann@ncl.ac.uk

**eLife digest** The arrival of sound waves at the ear causes the fluid inside a part of the ear known as the cochlea to vibrate. These vibrations are detected by tiny hair cells and transformed into electrical signals, which travel along the auditory nerve to the brain. After processing in the brainstem and other regions deep within the brain, the signals reach a region called the auditory cortex, where they undergo further processing.

The cells in the cochlea that respond to sounds of similar frequencies are grouped together, forming what is known as a tonotopic map. This also happens in the auditory cortex. However, the temporal properties of sounds—such as how quickly the volume of a sound changes over time—are represented differently. In the cochlea these properties are instead encoded by the rate at which the cochlear nerve fibres 'fire' (that is, the rate at which they generate electrical signals). However, it is not clear how the temporal properties of sound waves are represented in auditory cortex.

Baumann et al. have now addressed this question by scanning the brains of three awake macaque monkeys as the animals listened to bursts of white noise with varying properties. This revealed that just as neurons that respond to sounds of similar frequencies are grouped together within auditory cortex, so too are neurons that respond to sounds with similar temporal properties. When these temporal preferences are plotted on a map of auditory cortex, they form a series of concentric rings lying at right angles to the frequency map in certain areas.

Recent brain imaging studies in humans have also suggested the existence of a 'temporal map'. Further experiments are now required to determine exactly how neurons within the auditory cortex encode the temporal characteristics of sounds.

gradients for modulation rate and frequency have been shown, similar to those in the inferior colliculus. However, it is not clear how such an organisation might be preserved across the multiple auditory fields of primate cortex where different fields show different orientations of the tonotopic gradient. While earlier fMRI studies in humans (*Giraud et al., 2000*; *Schonwiesner and Zatorre, 2009*; *Overath et al., 2012*) reported robust responses to a range of amplitude-modulated sounds, but no systematic organisation of rate, two more recent studies suggested an orthogonal relationship of frequency and rate in areas homologous to non-human primate auditory core (*Herdener et al., 2013*) and beyond (*Barton et al., 2012*). Electrophysiology studies in non-human primates have shown tuning of individual neurons to different modulation rates and suggest a tendency for neurons in primary fields to prefer faster rates than neurons in field higher up the hierarchy (*Bieser and Muller-Preuss, 1996*; *Liang et al., 2002*): see also (*Joris et al., 2004*). However, no clear topographical organisation of modulation rate across different auditory fields has been demonstrated in non-human primates.

In the current study we mapped the blood oxygen level dependent (BOLD) response to a wide range of amplitude modulation rates from 0.5–512 Hz applied to a broad-band noise carrier in the auditory cortex of macaque monkeys using functional magnetic resonance imaging (fMRI). This range of modulation rates covers preferred rates for cortical neurons (*Joris et al., 2004*). We investigated whether the preference for specific amplitude modulation rates in neuronal ensembles is systematically represented in the auditory cortex and, if so, how such an organisation is arranged relative to the tonotopic gradients across auditory fields.

The data reveal a topographic organisation of amplitude modulation rate in the macaque auditory cortex arranged in concentric iso-rate bands that are mirror-symmetric across both hemispheres with a preference for the highest rates in the postero-medial auditory cortex at the medial border of A1 and for the lowest rates in lateral and anterior fields. This organisation results in a modulation rate gradient running approximately orthogonal to the tonotopic gradients in the auditory core fields, A1 and R.

## Results

### Amplitude modulation rate maps

In a first experiment we recorded the BOLD response to amplitude-modulated broad-band noise at six different rates (0.5, 2, 8, 32, 128, 512 Hz, see also [*Baumann et al., 2011*]) across the auditory cortex of three monkeys. We generated two different maps to reveal the spatial organisation of

modulation rate by projecting the data of the acquired volumes onto the cortical surface derived from the anatomical scans. In a first map (contrast map) we contrasted the response strength of the lower rate bands vs the higher rate bands to reveal the gradual change of preference for higher to lower modulation rates across the auditory cortex (*Figure 1*, top panels). The data corresponding to the two highest rates (512 Hz, 128 Hz) and lowest rates (2 Hz, 0.5 Hz) were combined before contrasting, while the intermediate rates were ignored (see also *Baumann et al., 2011*). In a second map (best-rate map), for each position in the auditory cortex, we represented the rate band that showed the strongest response. For the best-rate maps, all six individual rates were mapped (*Figure 1*, bottom panels). The contrast maps reveal a topography for different amplitude modulation rates with preferences for high rates consistently clustered in the postero-medial auditory cortex with the maxima at the medial border of the primary field A1 in both hemispheres of all tested animals. Preferences for low rates were located lateral, anterior and to some degree posterior to the high-rate clusters. In the areas with the maximal preference for low or high rates, the contrast between low and high rates was statistically significant ($p < 0.05$) in all animals and hemispheres, corrected for multiple comparisons with family-wise error (FWE) correction over the recorded volume. The best-modulation-rate maps for the six tested rates (*Figure 1*, bottom) confirmed the systematic organisation of the response pattern with a topographic representation of rate arranged in approximately concentric frequency bands starting with high rates in the postero-medial auditory cortex (at least a few voxels showed a best modulation rate of 128 Hz postero-medially in 5 of 6 hemispheres) and progressing anterior, lateral and in some cases posterior to lower rates (see also the schemata in Figure 4). The highest rate (512 Hz) was hardly represented in the best-rate maps.

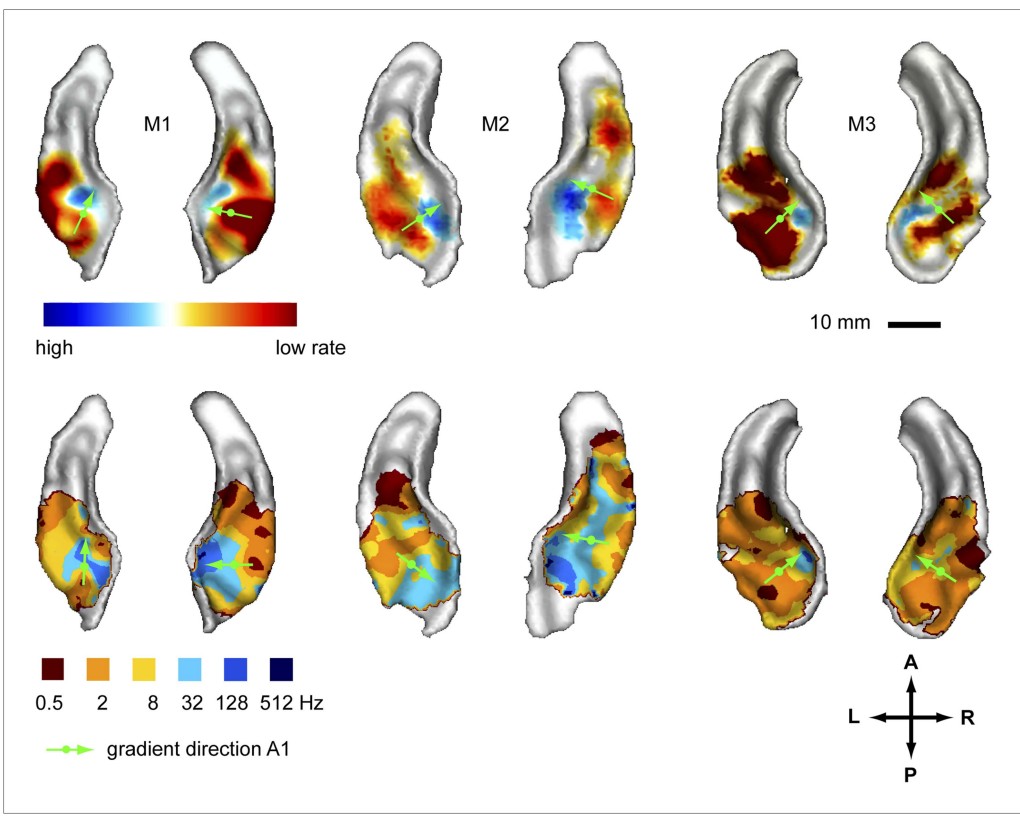

**Figure 1.** Representation of amplitude modulation rates in the auditory cortex. Top panels: map showing contrast of low vs high rates (rate contrast map) projected on rendered surfaces of the superior temporal planes in three animals (M1-3). Green arrows indicate mean gradient direction of the contrast in auditory field A1 (gradient directions derived from 2D regression; see also *Table 1*). Green circles indicate the position of the centre of mass of A1. Bottom panels: map of preferred response to different rates (Best-rate map). A; anterior, P; posterior, L; left, R; right. DOI: 10.7554/eLife.03256.003

## Frequency maps

In a second experiment, we recorded the BOLD response to bandpass noise in three different frequency bands (0.5–1 kHz, 2–4 kHz, 8–16 kHz) in the same animals used in the first experiment. Based on these data, we generated contrast maps (*Figure 2*, top panels) and best frequency maps (*Figure 2*, bottom panels) for each animal similar to experiment 1. These maps confirm the well-established tonotopic pattern with multiple reversals of the frequency gradient that serve as a basis for the delineation of auditory fields in the primate auditory cortex (*Morel et al., 1993*; *Kosaki et al., 1997*; *Petkov et al., 2006*; *Bendor and Wang, 2008*; *Baumann et al., 2010*). In a further experiment, we confirmed these findings using a 'phase-encoded' design (*Sereno et al., 1995*; *Joly et al., 2014*) in animal M1 and M2 (*Figure 3*). Based on these frequency maps, we identified auditory fields according to *Hackett (2011)*, using procedures described in *Petkov et al. (2006)* and *Baumann et al. (2010)*, adapted by the use of T1/T2 weighted MRI data to inform on the location of the core/belt border (*Joly et al., 2014*).

Here, we describe organisational features of these maps (summarised in *Figure 4*, right) which are well in line with previous studies (reviewed in *Baumann et al., 2013*). The anterior-posterior low to high gradient that defines the auditory core field A1 is particularly clear in the contrast maps (*Figure 2*, top panels). In line with previous studies in macaques, this gradient has generally a lateral-to-medial component in addition to the main anterior-posterior direction (*Morel et al., 1993*; *Kosaki et al., 1997*; *Baumann et al., 2010*). Similar to the situation in humans, the low frequency area that defines the border between fields A1 and R is typically wider on the lateral side compared to the medial side. The anterior high frequency area, which forms the border between the auditory fields R and RT, is typically located on the medial side of the superior temporal plane, in the depth of the

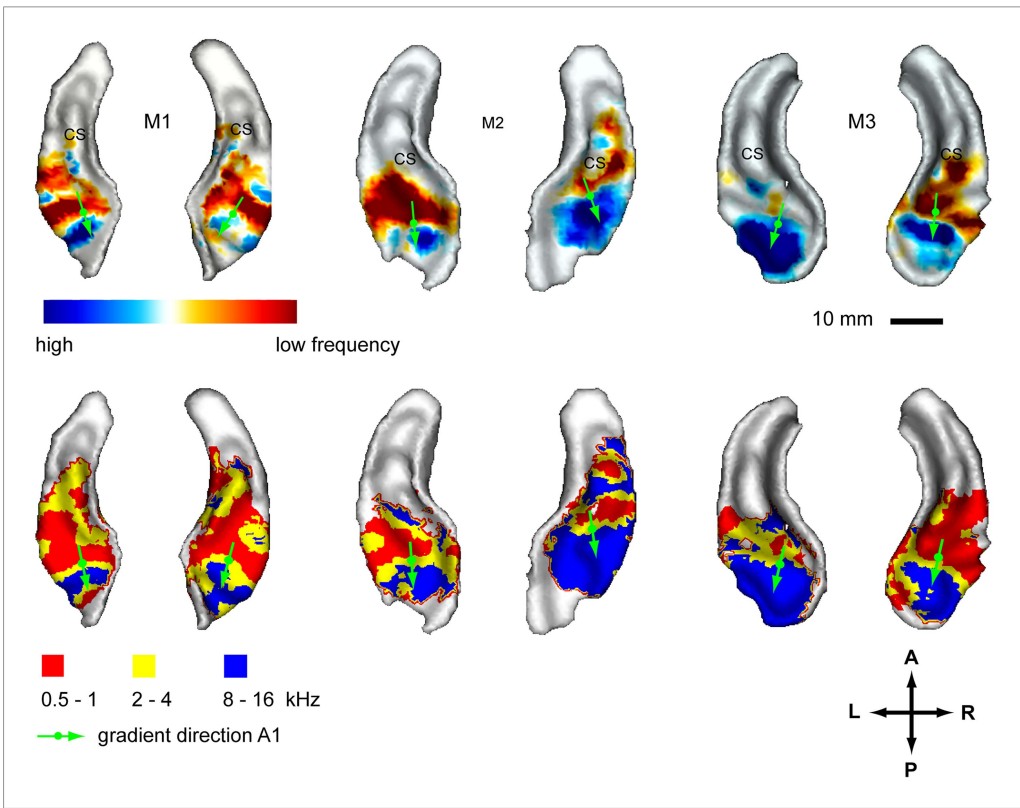

**Figure 2**. Representation of spectral frequency in the auditory cortex. Top panels: map of contrast of low vs high frequency band (frequency-contrast map) projected on rendered surfaces of the superior temporal planes in three animals (M1-3). Green arrows indicate mean gradient direction of the contrast in auditory field A1. Green circles indicate the position of the centre of mass of A1. Bottom panels: map of preferred response to different frequency bands (Best-frequency map). A; anterior, P; posterior, L; left, R; right.

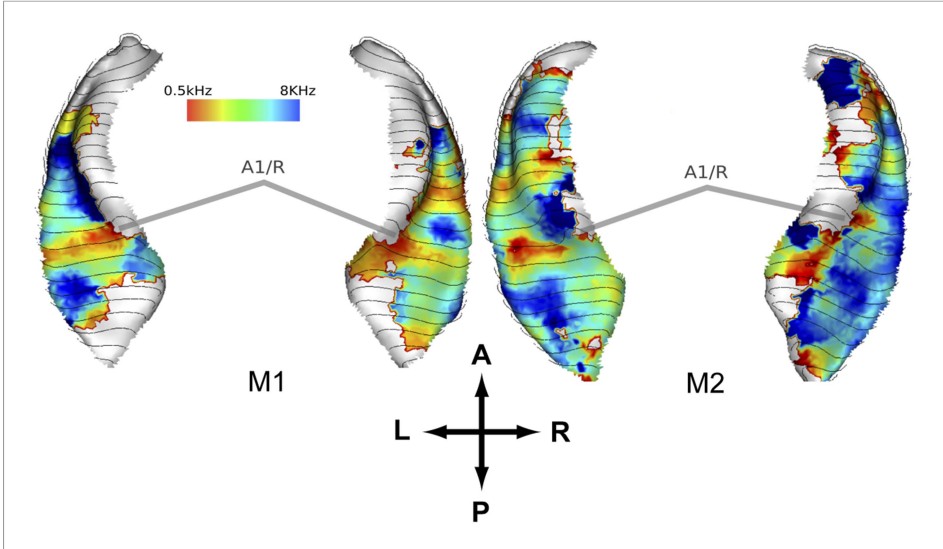

**Figure 3**. Representation of spectral frequency in the auditory cortex derived from 'phase-encoded' experiment. Top panels: map of preferred response to nine different frequencies between 0.5–8 KHz projected on rendered surfaces of the superior temporal planes in two animals (M1, M2). Approximate border between A1 and R is marked by grey bars. A; anterior, P; posterior, L; left, R; right.

circular sulcus (see also *Figure 4*, *Baumann et al., 2013*). The resulting anterio-medial direction of the main frequency gradient in R forms, in combination with gradient direction in A1, an inward inflection in the gradient axis across A1 and R, which has previously been reported in *Morel et al. (1993)* and *Kosaki et al. (1997)* in macaques, *Bendor and Wang (2008)* in marmosets and described in (*Jones, 2003*; *Hackett, 2011*; *Baumann et al., 2013*). This directional change of the frequency gradient axis across the core fields, which is also obvious from the different mean angles of the gradients in A1 and R (*Table 1*), is of particular relevance for the relationship of temporal and spectral gradients in these fields (see *Figure 4* and 'Discussion').

## Relative orientation of modulation rate and frequency gradients in auditory core

The systematic, topographic representation of amplitude modulation rate we demonstrate extends over multiple fields to form a wider concentric organisation across the entire auditory cortex in both hemispheres. If this organization is overlaid with the spectral pattern derived from experiment 2, we notice that in the auditory core areas (A1, R) the gradients for modulation rate and frequency lie approximately orthogonal to one another (illustrated in schemata of *Figure 4*). This arrangement is most obvious in A1 in individual animals, but the relationship generally holds for R and the adjacent lateral belt fields as well. This occurs as a consequence of the change in the direction of the tonotopic axis between A1 and R such that the tonotopic gradient follows approximately the iso- amplitude-modulation rate lines. In other words, the tonotopic axis runs anteriorly and medially in area R whilst the rate gradient runs anteriorly and laterally to preserve the orthogonal relationship. In this arrangement, the frequency reversals form spokes in the concentric organisation of the amplitude modulation rate (*Figures 1, 2, 4* left). However, rostral of R and caudal of A1 the orthogonality of spectral and temporal features breaks down and the gradients become more collinear and less obvious (*Figures 1, 2, 4* right).

In order to quantify these observations we calculated gradient directions and relative orientations based on the contrast maps for the topographies of modulation rate and frequency in the auditory core fields A1 and R using a two dimensional regression analysis as described in *Baumann et al. (2011)*. We also calculated the relative gradients for the caudio-lateral field CL, representative for extra-core areas where the orthogonal direction breaks down. The auditory fields have been delineated based on the tonotopy reversals as described in *Baumann et al. (2010)* and *Petkov et al.*

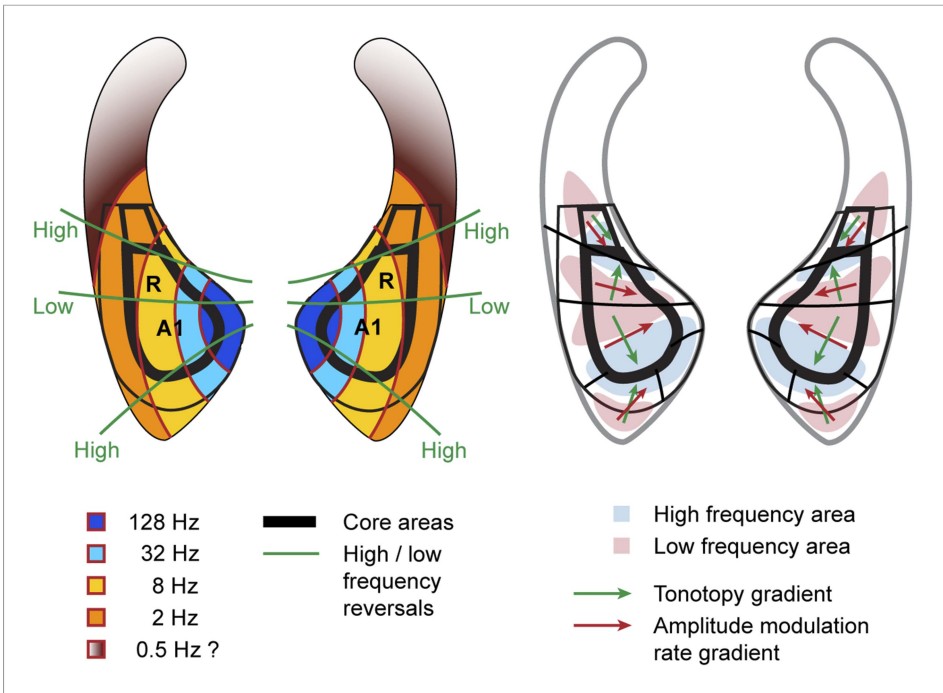

**Figure 4**. Schematic representation of amplitude-modulation-rate organisation in macaque auditory cortex. Model of modulation rate organisation in context of functional-field borders and frequency reversals (left side). Schematic organisation of tonotopy with indication of main gradients for tonotopy and modulation rate in selected functional fields (right side).

*(2006)*. The results are summarised in *Table 1* and gradient directions in A1 are indicated with green arrows in *Figures 1, 2* overlaid on the contrast maps with a circle marking the centre of mass of field A1. The gradient directions in the individual hemispheres generally follow the schemata in *Figure 4*. All the calculated fields show a clear gradient. In A1, the gradient directions for rate and frequency clearly cross each other but deviate somewhat from perfect orthogonality with an average angle of 118.7 ± 22.3°. The calculated directions in field R are more variable, but also show an average relative angle for rate and frequency gradients of about 120° (117.2 ± 47.6°). In contrast, relative angles in the postero-lateral field CL are much closer to anti-parallel with 4 of 6 hemispheres showing a relative angle around 160° and higher.

Due to their sparse, non-parametric nature, the best rate/frequency maps are less suited for gradient analysis. Furthermore, in some belt fields that feature a single best frequency, no gradient can be specified. However, for comparison, we provided a respective gradient analysis of the best rate/frequency for the fields with a defined gradient in the supplementary methods (*Supplementary file 1*) and the gradients for A1 are highlighted on the respective maps in *Figures 1, 2*. In most cases, the calculated gradient directions in core areas differ little from the analysis based on the contrast maps. The resulting relative angles are also similar with means closer to 110° in core areas (108.5 ± 42.1° for A1, 112.0 ± 55.1° for R). However, the correlation values ($r^2$), p values and the variance across animals and hemispheres are clearly worse, which can be attributed to the sparse nature of the data. This is particularly true for the posterior belt field CL.

## Discussion

Here we demonstrate for the first time a systematic and topographic representation of amplitude modulation rate in the auditory cortex of non-human primates. Concentric bands of decreasing rates extend bilaterally from the medial auditory cortex in anterior, lateral and posterior directions. In the region of the auditory core areas, the modulation gradients cross the well-established frequency gradients in approximately perpendicular direction leading to an orthogonal representation of the

**Table 1**. Directions and relative orientations of amplitude modulation rate and frequency gradients in selected auditory fields

| Animal | Hemis-phere | Field | Rel. angle (α) (degrees) | Frequency (degrees) | $R^2$ | p-value | Rate (degrees) | $R^2$ | p-value | N (vertices) |
|---|---|---|---|---|---|---|---|---|---|---|
| M1 | L | A1 | 137 | 162 | 0.796 | <1E-16 | 25 | 0.636 | <1E-16 | 84 |
|  | R |  | 73 | 155 | 0.546 | <1E-16 | 82 | 0.696 | <1E-16 | 101 |
| M2 | L |  | 120 | 175 | 0.927 | <1E-16 | 56 | 0.884 | <1E-16 | 154 |
|  | R |  | 133 | 160 | 0.803 | <1E-16 | 66 | 0.750 | <1E-16 | 103 |
| M3 | L |  | 121 | 164 | 0.824 | <1E-16 | 43 | 0.699 | <1E-16 | 156 |
|  | R |  | 128 | 175 | 0.722 | <1E-16 | 47 | 0.675 | <1E-16 | 87 |
| Average |  |  | 118.7 | 165.2 | 0.77 |  | 53.2 | 0.72 |  | 114.2 |
| Std dev |  |  | 23.3 | 8.2 | 0.13 |  | 19.7 | 0.09 |  | 32.5 |
| M1 | L | R | 81 | 38 | 0.106 | 1.40E-02 | 120 | 0.782 | <1E-16 | 88 |
|  | R |  | 140 | 23 | 0.391 | 2.89E-11 | 117 | 0.600 | <1E-16 | 107 |
| M2 | L |  | 51 | 29 | 0.697 | <1E-16 | 80 | 0.420 | 2.00E-09 | 102 |
|  | R |  | 103 | 44 | 0.441 | 2.50E-10 | 147 | 0.660 | <1E-16 | 93 |
| M3 | L |  | 180 | 81 | 0.476 | 1.40E-10 | 100 | 0.255 | 4.20E-06 | 87 |
|  | R |  | 148 | 9 | 0.611 | 4.20E-15 | 157 | 0.691 | <1E-16 | 73 |
| Average |  |  | 117.2 | 37.3 | 0.45 |  | 120.2 | 0.57 |  | 91.7 |
| Std dev |  |  | 47.6 | 24.6 | 0.20 |  | 28.7 | 0.20 |  | 12.1 |
| M1 | L | CL | 161 | 13 | 0.577 | 2.80E-04 | 148 | 0.587 | 1.70E-06 | 33 |
|  | R |  | 113 | 109 | 0.516 | 3.30E-07 | 138 | 0.264 | 1.90E-03 | 44 |
| M2 | L |  | 163 | 42 | 0.563 | 3.30E-07 | 155 | 0.582 | 2.00E-12 | 40 |
|  | R |  | 102 | 14 | 0.552 | 5.70E-10 | 116 | 0.559 | <1E-16 | 58 |
| M3 | L |  | 168 | 27 | 0.795 | 3.30E-16 | 165 | 0.751 | 2.60E-14 | 48 |
|  | R |  | 158 | 2 | 0.706 | 4.20E-11 | 156 | 0.768 | 4.20E-13 | 42 |
| Average |  |  | 144.2 | 34.5 | 0.62 |  | 146.3 | 0.59 |  | 44.2 |
| Std dev |  |  | 28.8 | 39.0 | 0.11 |  | 17.4 | 0.18 |  | 8.4 |

Main gradient directions (relative to anterior-posterior axis) and the resulting relative angle (α) between the orientations of the amplitude modulation rate (Rate) and spectral frequency (Frequency) gradients in auditory fields A1, R and CL are listed for two hemispheres (L, R) in three animals (M1-3). Additionally, $R^2$ values, p-values and number of data points (n) from the respective 2D regression analysis are included.

temporal and spectral dimensions of sound. This organisation increasingly breaks down in extra-core auditory fields that tend to show a preference for slow rates. The topographical maps for temporal modulations are largely symmetrical across the hemispheres showing no signs of a consistent lateralisation for temporal features.

The topographical representation of stimulus attributes is a common organising principle in the brain. Apart from sound frequency, we also find it in the multiple retinotopic representations in the visual system and the somatotopic representations in the somato-sensory system. Orthogonal representations of two topographic gradients have been previously demonstrated for other stimulus dimensions in the auditory system (*Suga and O'Neill, 1979*) and the visual system (e.g., *Tootell et al., 1982*; *Sereno et al., 1995*), and are predicted on theoretical considerations (*Swindale, 2004*; *Watkins et al., 2009*). In the current case the partial orthogonal representation might facilitate the simultaneous analysis of different dimensions of sound.

## Comparison of the results to previous studies in primates

Previous studies that investigated the representation of amplitude-modulation rate (*Bieser and Muller-Preuss, 1996*; *Giraud et al., 2000*; *Bendor and Wang, 2008*; *Schonwiesner and Zatorre, 2009*; *Overath et al., 2012*), have not reported a topographical organisation of this temporal

dimension in the auditory cortex. We can only speculate why such an organisation has not been reported in these previous studies. Electrophysiology studies in other primate species have highlighted a tendency for differential responses across different auditory fields with more primary areas preferring faster rates or showing shorter latencies and areas higher up the hierarchy preferring lower rates or showing longer latencies (*Bieser and Muller-Preuss, 1996*; *Bendor and Wang, 2008*). These results have been interpreted in terms of a posterior-to-anterior high-to-low rate gradient (*Bendor and Wang, 2008*) or a core-to-belt high-to-low rate gradient (*Camalier et al., 2012*). Given our results, both schemes have some merit, but they do not capture the full complexity of the organisation for temporal rates in concentric bands. Furthermore, the maps for amplitude-modulation rate reported in this study (*Figure 1*) indicate that in addition to differential responses between fields a clear gradient can be observed within the fields, particularly in the auditory core.

While earlier human fMRI studies (*Giraud et al., 2000*; *Schonwiesner and Zatorre, 2009*; *Overath et al., 2012*) did not show a clear topographical representation of amplitude modulation rate, two recent studies suggest a topographical gradient for this sound dimension (*Barton et al., 2012*; *Herdener et al., 2013*). Furthermore, both studies reported an orthogonal relationship of temporal rate and frequency in some auditory areas. The representation of modulation rates reported in *Barton et al. (2012)* in the human auditory cortex resembles the data from the current study with a maximal preference of high rates in the postero-medial cortex surrounded by areas with a preference for lower rates. However, the interpretation of the topographical pattern of this study differs to ours, in proposing a concentric organisation for frequency representation overlapping with an angular representation of modulation rate in contrast to our interpretation of a concentric organisation for modulation rate and an angular representation of frequency in core areas. An important result of this difference is that *Barton et al. (2012)* suggest an orthogonal relationship of frequency and modulation rate representation in the entire auditory cortex while in our case we find such a relationship mainly in the core areas. Furthermore, the study of Barton et al. suggested that each field contained a separate complete amplitude modulation rate gradient in addition to a separate and complete frequency gradient whilst the scheme that we demonstrate in macaques suggests that amplitude modulation rate is represented across multiple areas, none of which contain a complete map. However, a detailed comparison between the two studies is complicated by *Barton et al. (2012)*, use of a definition of the auditory fields that is incompatible with the definition commonly used in non-human primates.

The study by *Herdener et al. (2013)* focused on human homologues of the auditory core areas (hA1, hR) in the vicinity of the Heschl's gyrus. The reported results in these areas are consistent with ours in that they suggest orthogonal relationships of frequency and amplitude modulation rates with similar gradient directions found to those in our study. Finally, a further recent study in humans tested the representation of temporal and spectral features (*Santoro et al., 2014*). The chosen approach differed from our and previous studies in that a computational analysis was applied to test different models of stimulus feature representation. Thus, the emphasis was not on mapping individual rates and frequencies or identifying stimulus gradients. Furthermore, combined spectro-temporal modulations where used as stimuli in addition to pure temporal and spectral modulations. Nevertheless, the study identified an area in postero-medial auditory cortex, just posterior of the medial Heschl's gyrus, with a preference for fast temporal rates. Regions anterior and lateral of this area showed a preference for low temporal rates (summarised in Figure 7 of *Santoro et al., 2014*). Based on anatomical relationships in the auditory cortex between human and non-human primates as suggested in *Baumann et al. (2013)*, such a pattern for temporal rate preference is consistent with the concentric pattern of rate representation observed here.

## How does the BOLD response reflect sound modulation responses of single neurons?

In electrophysiology, neuronal responses to modulation rate are usually divided into two different coding principles: rate coding and temporal coding (e.g., *Lu et al., 2001*; *Liang et al., 2002*; *Yin et al., 2011*), reviewed in *Joris et al. (2004)*. The rate code is a measure of the average number of spikes fired over a defined period of time to the different modulation rates, and the temporal code is a measure of how well the neuron's firing synchronises with the amplitude envelope of the stimulus at different modulation rates. The average response of neurons in the cortex shows slightly different best

rates for the two measures and some neurons are properly tuned to only one of the two measures (*Joris et al., 2004*). Different neurons in the auditory cortex of non-human primates have been reported with tuning to modulation rates between 2–120 Hz, measured by temporal response and between 1–250 Hz measured by rate response (*Bieser and Muller-Preuss, 1996*; *Liang et al., 2002*), see also (*Joris et al., 2004*). However, neurons responding with a synchronised temporal response are rarely tuned to rates above 64 Hz and are preferentially located in primary areas. Neurons demonstrating rate tuning make up the majority of neurons that respond to modulated sounds in-non primary auditory areas and are frequently tuned to rates at 64 Hz and above.

It is not entirely clear which of the two response types are represented by the BOLD response reported in this study. While an increased average firing rate of a local sample of neurons to a certain modulation rate would certainly lead to an increased BOLD response, an increased synchronisation within the same sample of neurons would probably have a similar effect. This is supported by a simulation study which suggested that both types of firing patterns would influence the BOLD response in similar way (*Chawla et al., 1999*). In our study, the range of amplitude-modulation rates represented in the BOLD response and the pattern of areas that respond to specific rates is better matched by response properties reported for single neurons responding with a rate code in non-human primates (*Bieser and Muller-Preuss, 1996*; *Liang et al., 2002*; *Bendor and Wang, 2008*). Nevertheless, it is also possible that the measured BOLD signal is a response to a combination of rate- and temporal-coding neurons.

## Comparison of preferred amplitude modulation rates in humans and monkeys

The responses to different amplitude-modulation rates reported in this study are in line with results from previous electrophysiological studies in other non-human primates (*Bieser and Muller-Preuss, 1996*; *Liang et al., 2002*) in the range of the preferred rates (2–128 Hz) as well as the preferred rates for the different fields. The current study showed the highest preference in A1 for a rate of 32 Hz and to a lesser extent to 8 Hz while similar electrophysiology studies in monkeys additionally highlighted 16 Hz, a rate not used in this study. Human studies (*Giraud et al., 2000*; *Harms and Melcher, 2002*; *Overath et al., 2012*) showed slightly lower preferred rates between 2–8 Hz for primary areas while rates above 64 Hz where hardly represented. These species-specific values are consistent with psychophysical comparisons between humans and macaques in an amplitude modulation discrimination task showing average peak sensitivities at lower rates in humans (10–60 Hz) than in macaques (30–120 Hz) (*O'Connor et al., 2010*) or even a low-pass function in humans (*Viemeister, 1979*).

## Relevance of amplitude-modulation rate for other temporal response features

We demonstrated a topographic map for a specific temporal feature of sound, the rate of amplitude modulation. Temporal variation of sound can also be characterised by other means such as frequency modulation. Furthermore, differences in response latencies in areas at a similar hierarchy level were used as an indicator of the temporal resolution of the local circuits (*Langner et al., 1987*, *2002*). Various studies show however that different temporal response measures to different temporal stimulus features are highly correlated. *Liang et al. (2002)*, for example, shows a high correlation of single neuron responses in the auditory cortex of non-human primates to amplitude-modulation rate and frequency-modulation rate. Studies in the inferior colliculus showed good correlation of the representation of amplitude-modulation rate and the response latency of neurons in the inferior colliculus (*Langner et al., 1987*, *2002*). Furthermore, studies that recorded response latencies across different auditory fields, showed a tendency for longer latencies in rostral fields (*Bendor and Wang, 2008*; *Camalier et al., 2012*) and belt fields (*Camalier et al., 2012*), consistent with the preference for slower modulation rates we found in these areas. This suggests that preferred amplitude-modulation rate is representative of the processing, or integration, time windows which characterise the time scale over which temporal features are integrated in a particular area or circuit.

## Conclusions

Here, we demonstrate a systematic, topographical representation of modulation rate in the auditory cortex of a non-human primate, organised, in parts, orthogonally to the established gradient for

frequency. The systematic concentric organisation of amplitude-modulation rate that we demonstrate here, could provide an anatomical basis for the analysis of different modulation rates in separate modulation filterbanks (*Dau et al., 1997a*, *1997b*). The superposition of maps of modulation rate and frequency also occurs in the inferior colliculus (*Baumann et al., 2011*) where some higher modulation rates are represented than in the cortex, whilst in the cortex the modulation rates mapped decrease with greater distance from A1. The mapping of distinct dimensions of sound, amplitude-modulation rate and frequency, as distinct vectors in an anatomical space, is analogous to the mapping of polar angle and eccentricity that occurs at all processing levels of the visual system. Representation in the auditory cortex differs, however, in that a complete mapping of modulation rate does not occur within each cortical area, in contrast to the multiple and complete representations of spectral frequency that these areas contain.

## Materials and methods

### Experiment 1, 2

#### Animals
The data were obtained from three male macaque monkeys (Macaca mulatta) weighting 9–16 kg. Animals were implanted with a headholder under general anaesthesia and sterile conditions as described in detail previously (*Thiele et al., 2006*). Before scanning, the animals were habituated to the scanner environment. A custom-made primate chair was used to position the animal in the vertical bore of the scanner and head movements were minimised with a head holder. Details of the positioning procedures are given in (*Baumann et al., 2010*). All experiments were carried out in accordance with the UK, Animals (Scientific Procedures) Act (1986), European Communities Council Directive 1986 (86/609/EEC) and the US National Institutes of Health Guidelines for the Care and Use of Animals for Experimental Procedures, and were performed with great care to ensure the well-being of the animals.

#### Sound stimuli and presentation (see also *Baumann et al., 2011*)
Sound stimuli were created in MATLAB 7.1 (MathWorks, Natick, USA) with a sample rate of 44.1 kHz and 16 bit resolution. Stimuli for characterising the BOLD response to spectral frequencies were based on a random-phase noise carrier with three different pass-bands, 0.5–1 kHz, 2–4 kHz and 8–16 kHz resulting in three different stimuli that encompassed different spectral ranges. The carriers were amplitude modulated with a sinusoidal envelope of 90% depth at 10 Hz to achieve a robust response in the auditory system. The stimuli for characterising the temporal rates in the amplitude modulation experiment were also based on random-phase noise carrier but had a flat broad-band spectrum from 25 Hz to 16 kHz. This carrier was amplitude modulated at six different rates, 0.5 Hz, 2 Hz, 8 Hz, 32 Hz, 128 Hz and 512 Hz resulting in six different stimuli that covered a broad range of temporal rates identical to the data previously reported from the inferior colliculus (*Baumann et al., 2011*). The duration of all the stimuli was 6 s which included at least three cycles of the modulation in the case of the lowest temporal frequency. This duration is also sufficient for the BOLD response in the auditory cortex of macaques to reach a plateau (*Baumann et al., 2010*). The on- and off-set of the stimulus were smoothed by a linear ramp of 50 ms.

We presented the stimuli in the scanner at an RMS sound pressure level of 75 dB using custom adapted electrostatic headphones based on a Nordic NeuroLab system (NordicNeuroLab, Bergen, Norway). These headphones feature a flat frequency transfer function up to 16 kHz and are free from harmonic-distortion at the applied sound pressure level. Sound pressure levels were verified using an MR-compatible condenser microphone B&K Type 4189 (Bruel & Kjaer, Naerum, Denmark) connected by an extension cable to the sound level meter Type 2260 from the same company.

#### MRI hardware and imaging
Data were recorded in an actively shielded, vertical 4.7 T MRI scanner (Bruker Biospec 47/60 VAS) equipped with a Bruker GA-38S gradient system with an inner-bore diameter of 38 cm (Bruker BioSpin GmbH, Ettlingen, Germany). The applied RF transmitter-receiver coil (Bruker) was of a volume bird-cage design that covered the entire head of the animals. Functional and structural data were acquired from 2 mm thick slices that were aligned to the superior temporal plane and covered the temporal lobe. The slices were selected with the help of an additional structural brain scan in sagittal orientation.

## Functional scan parameters

Single-shot gradient-recalled echo-planar imaging sequences were optimised for each subject sharing an in-plane resolution of $1 \times 1$ mm$^2$ and a volume acquisition time (TA) of 1 s. Typical acquisition parameters were: TE: 21 ms, flip angle (FA): 90°, spectral bandwidth: 200 kHz, field of view (FOV): $9.6 \times 9.6$ cm$^2$, 16 slices of 2 mm thickness, with an acquisition matrix of $96 \times 96$. Each volume acquisition was separated by a 9 s gap to avoid recording the BOLD response to the gradient noise of the previous scan ('sparse design'). In combination with the TA of 1 s this results in a repetition time (TR) of 10 s. The stimuli were presented during the last 6 s of the silent gap. The detailed timing was based on a previous BOLD response time course characterisation in the auditory system of macaques (*Baumann et al., 2010*). Before every other volume acquisition stimuli were omitted to obtain data for a silent baseline. For the frequency experiment a total 720 vol were acquired per session. This resulted in 120 vol per stimulus per session (half of the volumes served for the baseline) or 360 vol per stimulus in total for the three sessions. For the amplitude modulation experiment 540 vol per session were acquired resulting in 45 vol per stimulus per session and 315 vol per stimulus for the combined seven sessions.

## Structural scan parameters

Structural images (T1-weighted) used the same geometry as the functional scans to simplify coregistration. The imaging parameters of the MDEFT (Modified Driven Equilibrium Fourier Transform) sequence were: TE: 6 ms, TR: 2240 ms, FA: 30°, FOV $9.6 \times 9.6$ cm$^2$ using an encoding matrix of $256 \times 256$ to result in an in-plane resolution of $0.375 \times 0.375$ mm$^2$ per voxel. Structural scans were acquired after each functional session.

## Data analysis

For preprocessing and general linear model analysis we employed the SPM5 software package (www.fil.ion.ucl.ac.uk/spm/) implemented in Matlab 7.1. The data acquired from each animal were analysed separately. Image volumes from each session were realigned to the first volume and the sessions of each experiment were subsequently realigned to each other before smoothing the data with a kernel of 2 mm full-width half-maximum. The time-series were high pass filtered with a cut-off of 300 s to account for slow signal drifts and the data was adjusted for global signal fluctuations (global scaling). In a general linear model analysis for the combined sessions of each experiment, the voxel-wise response estimate coefficients (beta-values) and t-values for the contrast of the different stimuli vs the silent baseline were calculated. Further analysis and data display was performed using custom designed Matlab scripts (*Source code 1*). The data were masked retaining only voxels that showed significant values for the combined stimuli vs baseline contrast for each of the two experiments (p < 0.001; uncorrected for multiple comparisons).

## Frequency/rate contrast maps

The frequency contrast maps were calculated voxel by voxel by subtracting the response estimate coefficients (beta-values) of the low frequency condition (0.5–1 kHz) from the high frequency condition (8–16 kHz). The rate contrast maps were calculated similarly, however the means of the lowest two rates (0.5, 2 Hz) and the highest two rates (128, 512 Hz) were taken before subtracting the response estimate coefficients; see also (*Baumann et al., 2011*). The resulting maps represent the degree of preference for high or low frequencies/rates.

## Best frequency/rate response maps (BF/R-map)

The best frequency/rate response maps were calculated by identifying voxel by voxel for each experiment and animal which of the three frequency conditions or six temporal rate conditions showed the highest t-values. The resulting maps represent the preferred frequency or rate for each voxel.

## Projection of data on anatomical surface

Structural images were segmented using ITKsnap (http://www.itksnap.org). The binary image was used to generate a 3D triangulated mesh of the superior temporal plane using BrainVisa suite (http://brainvisa.info). The data from of the Contrast- and Best-Frequency/rate Maps where then projected on the rendered surface using BrainVisa and taking, for each point on the surface, the data in a sphere of 1.6 mm into account.

## Analysis of gradients

For further analysis the data from the contrast maps on the surface was imported into Matlab as vertex mesh. Auditory fields according to (*Hackett, 2011*) were identified based on the gradient reversals of the tonotopic maps; see (*Baumann et al., 2010*) and (*Petkov et al., 2006*) for details, and auditory

core/belt borders suggested by myelinisation maps that were estimated from a T1/T2 contrast procedure as described in *Joly et al. (2014)*. For each auditory field the mean gradients of modulation rate and frequency preference were calculated in separate, two dimensional regression analyses as previously described in *Baumann et al. (2011)*. The input of the regression analysis was the spatial x and y coordinates and the values from the contrast maps, effectively resulting in gradients of a flat projection of the contrast values for the different auditory fields.

## Experiment 3

### Subjects

The two male rhesus monkeys that participated in this experiment were identical to subjects M1 and M2 in experiment 1 and 2. Before the scanning sessions, the monkeys were trained to perform a visual fixation task with the head of the animal rigidly positioned with a head holder attached to a cranial implant. The visual fixation task was used to equalise as much as possible attention across runs and minimize body movement and stress during scanning and presentation of the auditory stimuli.

### Stimuli

Sound stimuli were generated at the beginning of each functional run. The stimuli were computed with a sampling rate of 44.1 kHz using an in-house Python program—*PrimatePy*. PrimatePy mainly relies on Psychopy, a psychophysics package (www.psychopy.org/). Stimuli were pure tone bursts and were presented in either low-to-high or high-to-low progression of frequencies. Frequencies were 500, 707, 1000, 1414, 2000, 2828, 4000, 5657, and 8000 Hz (half-octave steps). Tone bursts were either 50 ms or 200 ms in duration (inter-stimulus interval 50 ms) and were alternated in pseudo-randomized order during the 2 s block, resulting in a rhythmic pattern of tone onsets. Pure-tone bursts of one frequency were presented for a 2 s block before stepping to the next frequency until all 9 frequencies had been presented. This 18 s progression was followed by a 12 s silent pause, and this 30 s cycle was presented 15 times. A run lasted for 8 min and the two run types with either low-to-high or high-to-low progression, were alternated. Stimuli were delivered through MR-compatible insert earphones (sensimetrics, Model S14, www.sens. com). Scan noise was attenuated by the earphones and by dense foam padding around the ears.

### Behavioural task

The animal performed the visual fixation task during the acquisition of a full time series (8 min). Each time series was followed by a break of about 1 min. The eye position was monitored at 60 Hz with a tracking (camera-based with Infra-Red illumination) of the pupil using iView software (SMI, www. smivision.com, Teltow, Germany). The task was as follow: a fixation target (a small red square) appeared on the centre of the screen, when the eye trace entered within a fixation window (about 2–3 visual degree centred onto the target) a timer started and the fixation target turned green. A continuous visual fixation (no saccades) of a randomly defined duration of 2–2.5 s was immediately followed by the delivery of a juice reward using a gravity-fed dispenser. The reward was controlled by PrimatePy via a data acquisition USB device LabJack (U3-LV, http://labjack.com/).

### Magnetic resonance imaging

Functional MRI measurements by blood oxygen level-dependent (BOLD) contrast consisted of single-shot gradient-echo echo-planar imaging sequences with the following parameters: TR = 1400 ms, TE = 21 ms, 90° flip angle, matrix 92 × 92, FOV 110 mm × 110 mm, in-plane resolution 1.2 × 1.2 mm$^2$, slice thickness = 1.2 mm. Functional times series consisted of a continuous acquisition of 343 vol with 20 axial slices acquired with parallel imaging with twofold GRAPPA acceleration using 8-channel array receive coil. The RF transmission was done with the Bruker birdcage volume coil in transmit mode. From the scanner, a TTL pulse signal was triggered at the start of every volume and sent out to PrimatePy via the LabJack for synchronization purposes. In total, a number of 15 runs were acquired (M1:8, M2:7) which represents 15 × 343 = 5145 vol.

Anatomical MR images consisted of two sequences, T1-weighted (T1w) and T2 weighted (T2w) images. The T1w images consisted of a 2D magnetization-prepared rapid gradient-echo (MPRAGE) sequence with a 130° preparation pulse, TR = 2100 ms, TE = 7 ms, TI = 800 ms, 27° flip angle. The T2w images consisted of a 2D Rapid Acquisition with Relaxation Enhancement (RARE) sequence with TR = 6500 ms, TE = 14 ms, RARE factor 8. The geometry was the same for both T1w and T2w images: matrix 166 × 166, FOV 100 mm × 100 mm, slices thickness 0.6 mm, and 54 axial slices. Because of time constraints, those anatomical scans were acquired during separate scanning sessions but with the same visual fixation task to minimise body motion and stress and to control the animal's behaviour.

## Data analyses

MR images were first converted from Bruker file format into 3D (anatomical data) or 4D (x, y, z, t functional data) minc file format (.mnc) using a Perl script pvconv.pl (http://pvconv.sourceforge.net/) and next from minc to nifti format using the *minc tools.*

Structural images were resampled at 0.25 mm isotropic voxels with seventh order B-spline interpolation method. Semi-automatic segmentation of the white matter was performed using ITKsnap (http://www.itksnap.org). The binary image (after dilation of 0.5 mm) was used to generate a 3D triangulated mesh (including smoothing) using BrainVisa suite (http://brainvisa.info) and a selection of the sub-surface corresponding to the Lateral Sulcus (LS) was saved into the GIFTI (www.nitrc.org/projects/gifti/) file format.

Raw fMRI data entered a preprocessing stage, including motion correction and spatial smoothing with a Gaussian kernel (FWHM = 1.5 mm). The time-series were further processed using python scripts (nitime and nibabel python libraries). Times series entered a filter with an infinite impulse response (IIR) function to remove fluctuations below 0.02 and above 0.1 Hz. The filtered times series of each voxel was then normalised as percentage of signal change relative to the mean signal of that voxel. For each voxel, cross-correlation between time-series from both run types was computed and time delay between the two signals (argument of the maximum correlation) revealed the preferred frequency. Volumetric preferred frequency maps were then projected onto the 3D cortical surface.

## Acknowledgements

The authors like to thank David Hunter for assistance with animal handling and Fabien Balezeau for assisting in data acquisition of experiment 3. The research was funded by the Wellcome Trust.

## Additional information

### Funding

| Funder | Grant reference | Author |
| --- | --- | --- |
| Wellcome Trust | WT 085002MA | Timothy D Griffiths |

The funder had no role in study design, data collection and interpretation, or the decision to submit the work for publication.

### Author contributions

SB, Conception and design of experiments, Acquisition of data for experiments 1 and 2, Analysis of experiments 1 and 2 Drafting of article; OJ, Conception and design of experiment 3, Acquisition of data for experiments 3, Analysis of experiments 3, Revising of article; AR, Conception and design of experiment 1, Revising of article; CIP, Contribution to analysis of experiments, Revising of article; LS, Provided MRI sequences and adjusted it for animals, Revised article; AT, Provided the animals and supervised their handling, Revised article; TDG, Conception and design of experiments 1 and 2, Drafting of article

### Author ORCIDs

Olivier Joly, http://orcid.org/0000-0002-5818-6552

### Ethics

Animal experimentation: All experiments were carried out in accordance with the UK, Animals (Scientific Procedures) Act (1986), European Communities Council Directive 1986 (86/609/EEC) and the US National Institutes of Health Guidelines for the Care and Use of Animals for Experimental Procedures, and were performed with great care to ensure the well-being of the animals.

## Additional files

### Supplementary files

• Supplementary file 1. Directions and relative orientations of amplitude modulation rate and frequency gradients in selected auditory field (based on best rate/frequency maps). Main gradient directions (relative to anterior-posterior axis) and the resulting relative angle (α) between the

orientations of the amplitude modulation rate (Rate) and spectral frequency (Frequency) gradients in auditory fields A1, R and CL are listed for two hemispheres (L, R) in three animals (M1-3). Additionally, $R^2$ values, p-values and number of data points (n) from the respective 2D regression analysis are included. * No defined gradient direction due to single best frequency in field.

• Source code 1. Custom matlab code. This file includes a collection of custom matlab scripts that were used for analysis and data representation of the manuscript. It is highly recommended to contact the corresponding author for detailed instructions in the use of these scripts and attempts to recreate analysis procedures described in the manuscript.

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
