## [Decision Letter]

Thank you for sending your work entitled “The Topography of Frequency and Time Representation in Primate Auditory Cortices” for consideration at *eLife*. Your article has been favorably evaluated by Eve Marder (Senior editor) and 2 reviewers.

In general, the reviewers agreed that the manuscript addresses an interesting question and that it offers important new data. Nonetheless, there are issues about how the data were analyzed that may call into question some of the conclusions, and that will need clarification before a final decision can be made (below).

The question of whether there is another representational gradient in auditory cortex—and what that representation might be—has been an ongoing debate in auditory neuroscience for some time, and has great relevance for models of audition and speech perception. Here the results of Baumann et al. suggest that amplitude modulation rate (varied from 0.5 to 512 Hz) is progressively mapped out over the superior temporal plane, with higher frequencies (∼128 or 32) typically evoking more activation in caudomedial cortex, with responses to progressively lower AM rates extending systematically out anteriorally and laterally. Not only are these results important for understanding non–human primate cortical auditory organization, with the first demonstration of 'amplitude–o–topy' in monkey but they also provide a vital cross–species link with, and confirmation of, a handful of very recent and somewhat controversial results in the human fMRI literature showing a similar spatial representation of AM. The tonotopic maps from the phase–encoded tonotopy experiment (resampled onto a reconstructed cortical surface) are also of extremely high quality, and extend our knowledge of the range of tonotopic mapping in macaques, and how it might relate to human maps similarly obtained.

Concerns:

There is a lot of emphasis on the direction of the tonotopic and AM gradients, and particularly that they are orthogonal to each other in primary auditory fields. This conclusion may rest on a number of analysis decisions.

First, rather than using data from the full range of stimulated frequencies for calculating gradients, the lowest two frequencies are subtracted from the highest, and the signed difference is plotted. On visual comparison of the full dataset (Figure 1, bottom; Figure 3) and high–low contrast maps (Figures 1 and 2, top), the latter appear to differ in the position/extent of the higher–frequency AM regions. For instance, the medial 'high' AM rate patch in M1 and M2 in the high minus low map has a more anterior center of gravity than in the 6–frequency map (which includes all of the data). This is particular pronounced in M1, where it considerably more anterior and lateral in the left hemisphere.

Second, the single illustrated gradient direction is strongly contingent upon the definition of the auditory field boundaries. The boundaries are based entirely on borders derived from a combination of the high–low tonotopy contrast maps with an a priori model of auditory field subdivisions based on the Hackett summary schematic. In looking at the figures, I suspect even slight changes to the shape and size of the borders would change the direction quite a lot. Also, it is not clear that the mean gradient is necessarily reflective of the local one if the isofrequency lines are highly curved, which they seem to be in several cases.

It would be much more straightforward for the reader to evaluate the authors' conclusions if in addition to the current figures, tonotopic and AM gradient maps were calculated for the full datasets (not just high–low, and including phase–encoded map), and shown for each subject/hemisphere. Because the maps are such high quality, the authors might also be able to calculate field sign with the combined maps (see Sereno, McDonald and Allman, 1994 for an exhaustive discussion of complex map depiction).

*Reviewer #1*:

In the Introduction section: “orthogonal topographical maps.” 'Orthogonal' is quite strong, 'non–parallel' or non–aligned would be more accurate I think.

In the Introduction it is strange (to say the least) not to cite the two published human fMRI studies that have tested amplitude modulation mapping (e.g., Barton et al., and Herdener et al.), especially when the following statement is made at the end of the second paragraph in the Introduction section: “However, as in humans, no clear topographical organisation of modulation rate across different auditory fields has been demonstrated in non–human primates.”

In the beginning of the Results section: Until later in the paper, it is unclear how auditory field borders are being identified. Even then it is unclear how A1/R/RT are defined given the multiple frequency reversals and bands evident in the more complete frequency maps from the 'phase–encoded' experiment.

Also in the beginning of the Results section: “This directional change of the frequency gradient axis across the core fields, which is often overlooked, is of particular relevance for the relationship of temporal and spectral gradients in these fields (see Discussion).” I found this section here and above difficult to follow because it heavily relies on co–reading the Baumann et al. review paper in Frontiers.

General methods: the AM rates span a very wide range, including ones (128 and 512Hz) that might induce pitch percepts (Burns and Viemeister, 1981, and others). The authors might want to discuss this possibility and how their results would or would not square with this interpretation. My feeling—particularly given the lack of apparent selectivity for 512Hz AM and the often smooth progression of isofrequency contours from 128Hz to 32Hz to 8 Hz—is that it is having a minimal effect if any, but other readers may have a different opinion.

*Reviewer #2*:

We are not asking you for new experiments to comply, but wish you to see it for your own edification.

“In my opinion, the weakest part of the paper is the discussion about the implications of the study. It would have been of great value to test the bold signal with some complex sounds, in particular monkey vocalizations. This is because the main hypothesis here is the existence of a functional map that represents periodic information that might be useful to preserve relevant behavioral cues. Even when mapping requires well controlled unidimensional variables, as in this paper, amplitude modulation rate might not necessarily be a mechanism for vocal communication. A proof of this is the fact that this cortical map was found without using complex sounds.”

---

## [Author Response]

*First, rather than using data from the full range of stimulated frequencies for calculating gradients, the lowest two frequencies are subtracted from the highest, and the signed difference is plotted. On visual comparison of the full dataset (*Figure 1*, bottom;*
Figure 3*) and high–low contrast maps (*Figures 1 and 2*, top), the latter appear to differ in the position/extent of the higher–frequency AM regions. For instance, the medial 'high' AM rate patch in M1 and M2 in the high minus low map has a more anterior center of gravity than in the 6–frequency map (which includes all of the data). This is particular pronounced in M1, where it considerably more anterior and lateral in the left hemisphere*.

We appreciate the specific interest in the “best rate/frequency” maps, and the gradient calculations that are derived from this method of representation. These maps provide additional, and to some degree complementary, information on the local preference to the mapped stimulus features compared to contrast maps. Furthermore, the “best frequency” approach is common in previous literature, and is the best approach to demonstrate tonotopic mapping in neurons with narrow tuning similar to that in the ascending pathway. For these reasons we provided both types of maps in the previously submitted version of the article and we are happy to provide an additional gradient analysis for the best frequency/rate approach in the current manuscript. As we highlighted in the modified text of the manuscript (last paragraph in the Results section), the results of this additional analysis generally agrees with the previous analysis based on contrast maps (see also [Supplementary-material SD1-data]).

However, we feel it is important to counter the impression that the best frequency/rate approach is more precise or more accurate than the contrast approach. In fact, our own experience, supported by increasing evidence from independent studies, suggests the opposite when the response of neuronal mass activity is mapped as in the case of the BOLD effect. Data from tonotopy studies show that at the level of MRI voxels containing 100,000 neurons the tuning curve is relatively broad and the peak response difference for distinct frequencies, particularly in the area of the middle frequencies, is small while the differences in the slopes of the tuning curves in more extreme frequency values is preserved (see also Langers et al., 2014). The reason for this effect is probably the heterogeneity of local frequency tuning of neighbouring neurons, which is particularly pronounced in middle frequencies (see Aschauer et al., 2014 for work in rodent: although a different species the basic phenomenon of tonotopic mapping is remarkably preserved across mammalian species and a similar organisation would explain data such as those of Langers). In the previously submitted iteration of this manuscript to *eLife* we have shown that the response to amplitude modulation rate shows similarly broad tuning curves. Thus, the contrast maps for frequency and amplitude modulation rate are more robust and show less inter-individual variability than the best frequency responses.

In the case of the gradient analysis the best frequency/rate approach has a further disadvantage. While the contrast response provides a continuous distribution of preference values, the best frequency/rate maps only contain six discreet values. Hence, a regression analysis is inherently less precise, particularly in areas, such as the belt, where the response is dominated by a single frequency/rate. This leads to lower correlation and less significance of the individual gradients and higher variability across subjects. For these reasons we decided to provide the results for the best frequency/rate gradient analysis in [Supplementary-material SD1-data] and not in the main section of the manuscript. However, the main gradients for A1 are displayed on the best frequency/rate maps in Figures 1 and 2 similar to the gradients for the contrast maps.

Additionally, we added the following text to the end of the Results part

“Due to their sparse, non-parametric nature, the best rate/frequency maps are less suited for gradient analysis. Furthermore, in some belt fields that feature a single best frequency, no gradient can be specified. However, for comparison, we provided a respective gradient analysis of the best rate/frequency for the fields with a defined gradient in [Supplementary-material SD1-data] and the gradients for A1 are highlighted on the respective maps in Figure 1 and Figure 2. In most cases, the calculated gradient directions in core areas differ little from the analysis based on the contrast maps. The resulting relative angles are also similar with means closer to 110° in core areas (108.5 ± 42.1° for A1, 112.0 ± 55.1° for R). However, the correlation values (r2), p values and the variance across animals and hemispheres are clearly worse, which can be attributed to the sparse nature of the data. This is particularly true for the posterior belt field CL.”

*Second, the single illustrated gradient direction is strongly contingent upon the definition of the auditory field boundaries. The boundaries are based entirely on borders derived from a combination of the high–low tonotopy contrast maps with an a priori model of auditory field subdivisions based on the Hackett summary schematic. In looking at the figures, I suspect even slight changes to the shape and size of the borders would change the direction quite a lot. Also, it is not clear that the mean gradient is necessarily reflective of the local one if the isofrequency lines are highly curved, which they seem to be in several cases*.

The main message we convey in this article is the discovery of a systematic, concentric organisation of amplitude modulation rate and its relationship to the previously identified tonotopy gradient. This finding is most obvious from the repeated pattern across two hemispheres and three animals in the maps provided in Figures 1 and 2. However, this finding is difficult to quantify as a whole. Thus we decided to test the general pattern and its relationship to tonotopy in generally accepted auditory fields for which the suggested scheme provides precise, testable hypotheses.

We believe that the applied method to define the borders of these widely accepted auditory fields is the best that is currently available. We apologise for not updating our methods paper describing the technique in macaques that is now published ([18]; Frontiers in Neuroscience): the areal mapping distinguishes core and belt based on T1/T2 ratio. While we do not deny that the definition of the borders has some influence on the precise direction of the gradients in these fields, such a change would have no influence on the gradual representation of decreasing amplitude modulation rate as you move away from A1 (the most striking finding in this study) nor its general relationship to the tonotopy gradient.

Reviewer #1:

*In the Introduction section:* “*orthogonal topographical maps.*” *'Orthogonal' is quite strong, 'non–parallel' or non–aligned would be more accurate I think*.

We have qualified the description in the manuscript (e.g. “approximately orthogonal”). However, we prefer orthogonal to non-parallel or non-aligned because: 1) it not only refers to a directional relationship but also to the idea that this process maps distinct stimulus dimensions irrespective of the exact relative gradient direction, and 2) non-parallel or non-aligned could mean anything but parallel (e.g. could refer to both a ten degree and a five degree relative angle), and is thus not very helpful in describing the finding. While we admit that the identified gradients in the core areas are not precisely perpendicular, this could neither be expected in a biological sample nor would it be necessary to be functionally relevant.

Incidentally, the expression in the Introduction section that the reviewer highlighted is a direct citation that refers to two previous articles.

*In the Introduction it is strange (to say the least) not to cite the two published human fMRI studies that have tested amplitude modulation mapping (e.g., Barton et al., and Herdener et al.), especially when the following statement is made at the end of the second paragraph in the Introduction section:* “ *However, as in humans, no clear topographical organisation of modulation rate across different auditory fields has been demonstrated in non–human primates*.”

We agree and thank the reviewer for improving the balance of our exposition. We now also include citations to the two mentioned studies in the Introduction, in addition to the subsequent text as previously: “While earlier fMRI studies in humans (14; 34; 29) reported robust responses to a range of amplitude modulated sounds, but no systematic organisation of rate, two more recent studies suggested an orthogonal relationship of frequency and rate in areas homologue to non-human primate auditory core (17) and beyond (1). Electrophysiology studies in non-human primates have shown tuning of individual neurons to different modulation rates and suggest a tendency for neurons in primary fields to prefer faster rates than neurons in field higher up the hierarchy (6; 25): see also (20). However, no clear topographical organisation of modulation rate across different auditory fields has been demonstrated in non-human primates.”

*In the beginning of the Results section: Until later in the paper, it is unclear how auditory field borders are being identified. Even then it is unclear how A1/R/RT are defined given the multiple frequency reversals and bands evident in the more complete frequency maps from the 'phase–encoded' experiment*.

At this point, it was intended to refer to a methods paper (18) that was about to appear at the time of the resubmission. We failed to update the manuscript when the paper with a detailed description of the applied procedure was published just before the resubmission. A brief description of the adaption of the previous procedure and the respective reference are now included in the revised version in the Results and in the Methods.

*Also in the beginning of the Results section:* “*This directional change of the frequency gradient axis across the core fields, which is often overlooked, is of particular relevance for the relationship of temporal and spectral gradients in these fields (see Discussion).*” *I found this section here and above difficult to follow because it heavily relies on co–reading the Baumann et al. review paper in Frontiers*.

We agree that the organisational features that are highlighted in this paragraph are difficult to appreciate without referring to recent findings that are summarised in [4]. A considerable number of recent high resolution imaging studies that describe tonotopic maps in human and non-human primates provide a more complete but also a more complex view of the tonotopic organisation in primates than previous schemes. In the previous version of the manuscript, we provided a summary of these results in a schematic representation in Figure 4 to avoid the necessity to consult the mentioned review and other cited literature but for validation of provided schemata. By slightly rearranging the paragraph and by highlighting the schemata in Figure 4 at the beginning we hope to improve the structure of this paragraph.

*General methods: the AM rates span a very wide range, including ones (128 and 512Hz) that might induce pitch percepts (**Burns and Viemeister, 1981**, and others). The authors might want to discuss this possibility and how their results would or would not square with this interpretation. My feeling—particularly given the lack of apparent selectivity for 512Hz AM and the often smooth progression of isofrequency contours from 128Hz to 32Hz to 8 Hz—is that it is having a minimal effect if any, but other readers may have a different opinion*.

It is true that stimuli with periodicities above about 30 Hz onwards are the pitch range of humans as well as of macaques as we recently showed behaviourally ([18]; Frontiers in Perception Science). However, the AM stimuli with noise carrier that we used in this study only elicit a very weak pitch percept. Furthermore, we are currently preparing a manuscript that is describing the response to more salient pitch stimuli with similar rates in the auditory cortex of macaques. Interestingly, the response pattern is considerably different in that case, providing further evidence that the pattern described here is not due to pitch. The response to periodic stimuli with rates above the lower limit of pitch (∼30 HZ) occurs in a region abutting but outside of A1 (see also Griffiths and Hall, J Neurosci, 2012 for case of regular interval noise—we have examined harmonic stimuli since), which is a different pattern to the one here where high rates are represented within A1.

Reviewer #2:

We are not asking you for new experiments to comply, but wish you to see it for your own edification.

“*In my opinion, the weakest part of the paper is the discussion about the implications of the study. It would have been of great value to test the bold signal with some complex sounds, in particular monkey vocalizations. This is because the main hypothesis here is the existence of a functional map that represents periodic information that might be useful to preserve relevant behavioral cues. Even when mapping requires well controlled unidimensional variables, as in this paper, amplitude modulation rate might not necessarily be a mechanism for vocal communication. A proof of this is the fact that this cortical map was found without using complex sounds*.”

We completely agree that the processing of amplitude modulation is highly relevant for animal vocalisations and human speech, as has been highlighted a several behavioural studies. However, at the current stage, the purpose of this study was simply to test whether this temporal dimension is represented in a systematic topographical organisation in the monkey auditory cortex. However, we do not deny that the results presented here could be further used in various ways to look into the interaction of this representation with behavioural results from monkey vocalisations.